# A Triple Gene-Deleted Pseudorabies Virus-Vectored Subunit PCV2b and CSFV Vaccine Protect Pigs against a Virulent CSFV Challenge

**DOI:** 10.3390/v15112143

**Published:** 2023-10-25

**Authors:** Ediane Silva, Elizabeth Medina-Ramirez, Selvaraj Pavulraj, Douglas P. Gladue, Manuel Borca, Shafiqul I. Chowdhury

**Affiliations:** 1US Department of Agricultural, ARS, Plum Island Animal Disease Center, Orient, NY 11957, USA; ediane.silva@usda.gov (E.S.); elizabeth.ramirez@usda.gov (E.M.-R.); douglas.gladue@usda.gov (D.P.G.); manuel.borca@usda.gov (M.B.); 2Department of Pathobiological Sciences, School of Veterinary Medicine, Louisiana State University, Baton Rouge, LA 70803, USA; pselvaraj1@lsu.edu

**Keywords:** pseudorabies virus, triple mutant, vectored vaccine, PCV2 capsid, classical swine fever virus (CSFV), glycoproteins E2 and E^rns^, PRV trivalent vaccine, vaccine efficacy, granulocytic monocyte-colony stimulating factor (GM-CSF), pig, CSFV challenge, subunit vaccine

## Abstract

Classical swine fever (CSF) remains one of the most economically significant viral diseases affecting domestic pigs and wild boars worldwide. To develop a safe and effective vaccine against CSF, we have constructed a triple gene-deleted pseudorabies virus (PRVtmv)-vectored bivalent subunit vaccine against porcine circovirus type 2b (PCV2b) and CSFV (PRVtmv+). In this study, we determined the protective efficacy of the PRVtmv+ against virulent CSFV challenge in pigs. The results revealed that the sham-vaccinated control group pigs developed severe CSFV-specific clinical signs characterized by pyrexia and diarrhea, and became moribund on or before the seventh day post challenge (dpc). However, the PRVtmv+-vaccinated pigs survived until the day of euthanasia at 21 dpc. A few vaccinated pigs showed transient diarrhea but recovered within a day or two. One pig had a low-grade fever for a day but recovered. The sham-vaccinated control group pigs had a high level of viremia, severe lymphocytopenia, and thrombocytopenia. In contrast, the vaccinated pigs had a low–moderate degree of lymphocytopenia and thrombocytopenia on four dpc, but recovered by seven dpc. Based on the gross pathology, none of the vaccinated pigs had any CSFV-specific lesions. Therefore, our results demonstrated that the PRVtmv+ vaccinated pigs are protected against virulent CSFV challenge.

## 1. Introduction

Classical swine fever (CSF) is a highly contagious and economically significant viral disease affecting pigs. The disease is endemic in China, Southeast Asian, South and Central American, and East and Central European countries [1,2].

The reintroduction of the virus in disease-free areas can be devastating. In 1997–1998, an outbreak in the Netherlands spread to involve more than 400 herds and cost $2.3 billion to eradicate. Approximately 12 million pigs were killed, some in eradication efforts but most for welfare reasons associated with the epidemic [3,4,5,6]. North America is also at risk of accidentally reintroducing CSF because CSF (CSFV) is still endemic in South and Central America and Caribbean countries [7]. Even though CSFV is mainly transmitted by direct contact with CSFV-infected animals, the indirect transmission of the virus through contaminated feed or swill brought by international travelers to the CSFV-free countries could also be a potential source. 

To prevent CSF losses, routine CSFV live modified/attenuated virus (MLV) vaccination is practiced in pigs of many affected regions (e.g., China, Russia, and Southeast Asia) [8]. The most widely used vaccine is the well-known lapinized “Chinese” C-strain-based MLV. The “C”-strain MLV [9] is known for inducing early protection in pigs against the virulent CSFV challenge at 3–5 days post intranasal vaccination [10]. The vaccine has been in use in China for more than two decades. Since the “C”-strain vaccine virus does not have a serological marker, the vaccine origin of the sporadic CSFV outbreaks cannot be ruled out [11]. Due to this limitation, European Union (EU) countries do not permit the vaccination of pigs with the otherwise highly effective MLV “C”-strain or similar MLV vaccines. Further, there is a restriction on trade from countries using the MLV CSFV or that are not yet CSFV-free, which has enormous economic consequences. 

Recently, the European Medical Agency approved a chimeric DIVA (differentiating infected from vaccinated animals) marker vaccine (Suvaxyn CSF Marker; Zoetis Belgium SA, Belgium) for use as an emergency vaccine in pigs only in the case of a CSFV outbreak. The chimeric vaccine is based on a bovine viral diarrhea virus (BVDV, cytopathic strain “CP7”) [12] expressing E2 envelope glycoprotein of CSFV strain, Alfort/187 [13]. Although the “CP7_E2alf” vaccine is safe in calves under experimental conditions [14], its safety and stability under field conditions have not been evaluated. 

Domestic pigs in North America, the EU, Australia, and New Zealand are pseudorabies virus (PRV)-free [15,16,17,18]. However, the virus remains endemic in feral pigs/wild boars in most PRV-free countries, including North America [19,20,21]. PRV is endemic in the domestic pig populations of Asia, including China, eastern and southeastern Europe, and Latin America [17,18,22]. Therefore, there is always a risk of reintroducing PRV in domestic pigs, even in PRV-free countries, which may cause economic losses. 

The co-infection of PRV with other viruses such as CSFV, porcine circovirus type 2 (PCV2), and porcine reproductive and respiratory syndrome virus (PRRSV) are very common in most of the pork-producing countries in eastern Europe and Asia, including China. A study in China revealed that the probability of PRV co-infection with PRRSV, PCV2, and CSFV was 36%, 12.9%, and 1.8%, respectively [23]. Therefore, the complex epidemiological interaction between the circulating PRV with other co-infecting viruses might have resulted in the emergence of the highly neurovirulent PRV strains among the Bartha K61 immunized pigs in China [23,24,25,26]. 

To develop a safer and more effective virally vectored multivalent vaccine against PRV, CSFV, and PCV2, we constructed a novel triple gene-deleted PRV subunit vaccine vector (PRVtmv), in which PCV2b capsid (Cap) and CSFV envelope proteins E2 and E^rns^-GMCSF (granulocyte macrophage colony-stimulating factor) are incorporated (PRVtmv+) [27]. Recently, we reported that the PRVtmv+ vaccine was effective against PCV2 challenge in pigs by preventing the viremia and PCV2-mediated leukopenia and immunosuppression [27]. 

Since we hypothesized that the PRVtmv+ would also be protective against CSFV, in this study, we validated our hypothesis by performing the vaccination challenge experiment in pigs, demonstrating that PRVtmv+ vaccinated pigs are protected against the fatal CSFV clinical disease and gross lesions associated with CSFV by inducing a moderate level of virus-neutralizing antibodies and significantly reducing the viremia. 

## 2. Materials and Methods

### 2.1. Cells, Medium, Virus, and Titration

The PRVtmv+ vaccine virus was propagated in swine kidney cells (SK) and titrated in Madin-Darby bovine kidney cells (MDBK), as reported earlier [27]. SK and MDBK cells were grown in 10% heat-inactivated fetal bovine serum (FBS; Equa1FETAL, Atlas Biologicals, Fort Collins, CO, USA) and 1× antibiotic-antimycotic solution (34-004-CI, Corning^®^, Corning, NY, USA) supplemented with Dulbecco’s modified Eagle’s medium (DMEM; #10-017-CV, Corning^®^, Corning, NY, USA). The CSFV Brescia strain (BICv) was derived from full-length cDNA copies and propagated in SK cells in DMEM medium [28,29]. CSFV titration was performed using SK cells in 96-well plates, as described earlier [30]. Briefly, the viral stock was serially diluted ten-fold in DMEM. A fixed volume of virus-dilution was applied in duplicate into the wells of 96-well cell culture plates over confluent SK cells. The plates were incubated for 96 h at 37 °C. After 4 days in the culture, viral infectivity was determined with an immunoperoxidase assay using the CSFV monoclonal antibody (mAb) WH174 or mAb WH303. The virus titer was calculated using Reed and Muench method [31], and virus titer is expressed as TCID_50_/_mL_. 

### 2.2. Viruses

Construction and characterization of PRVtmv+ vaccine vector have been reported earlier [27]. A virus derived from an infectious clone encoding the CSFV Brescia strain (BICv) was used as a challenge virus [28].

### 2.3. Animals 

The animal experiment was performed under biosafety level 3 conditions in the Plum Island Animal Disease Center animal facilities, following a strict protocol approved by the Institutional Animal Care and Use Committee (Number 171.12-21-R, approved on 12-09-21). Further, the LSU Institutional Animal Care and Use Committee endorsed the above-approved protocol. 

Based on their vaccination records, all pigs were vaccinated against porcine reproductive and respiratory syndrome virus (PRRSV), PCV2, swine influenza, *Hemophilus parasuis*, and *Mycoplasma hyopneumoniae*. Further, the pigs were clinically healthy at the time of immunization. Ten 30–40 lbs. (five-week-old) healthy female Yorkshire crossbred pigs were divided randomly into two groups of five pigs each, and housed in two separate rooms. For randomization, ten pigs were assigned with numbers from 83 to 92. Subsequently, an online tool was used to randomly shuffle the order of numbers and assign the subjects into two groups, viz. CSFV control and PRVtmv+ vaccine group. 

### 2.4. Vaccination and Challenge 

The vaccination, CSFV challenge, and sample collection schemes are shown in Figure 1A. After acclimatizing for seven days, group 1 (control group) and group 2 (PRVtmv+ vaccine group) pigs were sham-vaccinated (cell culture media) or vaccinated with the prototype PRVtmv+ vaccine, using both intranasal (IN) and subcutaneous (SC) vaccination routes, as described earlier [27]. Briefly, 4 × 10^7^ plaque-forming units (PFUs) were used per nostril (total 8 × 10^7^) and filtered (0.2 μm pore size) 4 × 10^7^ PFUs were inoculated intranasally and injected subcutaneously, respectively. At 28 days post vaccination (dpv), all pigs (sham-vaccinated control and vaccinated) were challenged intranasally with 10^5^ TCID_50,_ of the challenge virus BICv.

#### 2.4.1. Clinical Examination and Sample Collection from Pigs Following Vaccination and Challenge

Pigs were monitored daily for 28 days after vaccination and for 21 days following the BICv challenge for any visible clinical signs. Especially after the challenge, clinical signs associated with the CSFV disease, i.e., inappetence, depression, fever, purple skin discoloration, staggering gait, diarrhea, and cough, were recorded. Blood samples were collected from the anterior vena cava for whole blood and serum on 0, 4, 7, 14, and 21 days post challenge (dpc). Pigs showing severe CSF clinical signs were euthanized. All pigs surviving the CSFV challenge were euthanized 21 dpc. A complete necropsy was performed, and gross lesions were recorded.

#### 2.4.2. Total and Differential White Blood Cell Counts 

Blood was obtained from the anterior vena cava in ethylenediaminetetraacetic acid (EDTA)-containing tubes (Vacutainer). Total white blood cell, lymphocyte, and platelet counts were obtained using a Beckman Coulter AcT (Beckman, Coulter, CA, USA). 

### 2.5. Detection of CSFV Viremia

To detect viremia, whole EDTA blood was serially diluted ten-fold, and 100 μL from each dilution was inoculated in 96-well plates (Costar, Cambridge, MA, USA), followed by adding SK cells (1 × 10^4^ cells per well). Plates were incubated at 37 °C and 5% CO_2_ for four days. After four days of incubation, the supernatant was removed from each well, and the cells were fixed with methanol–acetone (50% *v*/*v*) solution and air-dried. The presence of virus-infected cells was determined by staining monolayers, as described earlier, with CSFV E2-specific monoclonal antibody WH303 in immunoperoxidase assay using the Vecstatin ABC Kit (Vector Laboratories, Burlingame, CA, USA), following the manufacturer’s instructions [32]. Titers were calculated and expressed as TCID_50_/_mL_, as described previously, with a sensitivity of detection of ≥1.8 TCID_50_/_mL_ [28]. 

**Figure 1 viruses-15-02143-f001:**
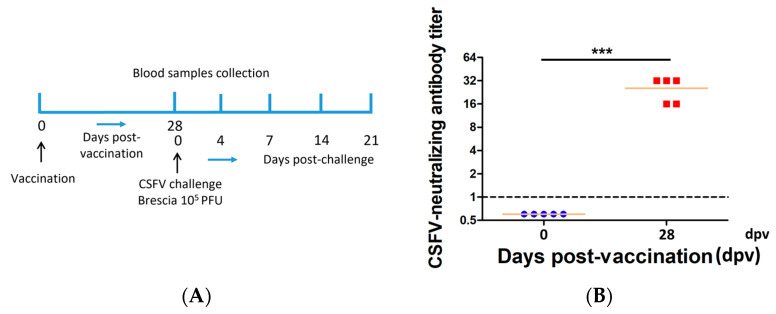
PRVtmv+ immunization in pigs. (**A**) Schematic showing the PRVtmv+ vaccination, sample collection, and CSFV challenge scheme for the animal experiment. (**B**) Pigs (*n* = 5 in each group) were immunized with PRVtmv+ vaccine (for each pig, intranasally—8 × 10^7^ plaque forming units (PFU), and subcutaneously—4 × 10^7^ PFU; 0 days post vaccination (dpv)), or sham-vaccinated and sera samples were collected on 0 and 28 dpv to determine the classical swine fever virus (CSFV)-specific neutralizing antibody titer [33]. The dot plot graph shows each animal’s individual CSFV titer with the mean values of the group (*n* = 5). *** *p* < 0.001.

### 2.6. Detection of Neutralizing Antibodies 

Serum neutralization assays were performed with heat-inactivated serum samples (56 °C for 30 min) [33]. Two-fold serial dilutions of serum were prepared in Dulbecco’s modified Eagle’s medium (DMEM) supplemented with 10% FBS and mixed with equal volumes of BICv containing 10^2^ TCID_50_. Serum–virus mixtures were incubated for one hour at 37 °C and then transferred to 96-well flat-bottom tissue culture plates (Corning), followed by adding SK cells (1 × 10^4^ cells per well). Plates were incubated at 37 °C and 5% CO_2_ for 4 days. As mentioned above, after four days, the supernatant was removed, and the cells in 96-well plates were processed for immunohistochemistry using CSFV E2-specific mAb. Neutralizing antibody titers were expressed as the reciprocal of the highest serum dilution (two-fold) that neutralizes BICv, as described elsewhere [31].

### 2.7. Statistical Analysis 

Data were analyzed for statistical significance (*p* < 0.05) with GraphPad prism 5.04 (San Diego, CA, USA). Comparisons of mean CSFV-specific neutralizing antibody titer and CSFV viremia in blood were performed using a two-way analysis of variance (ANOVA) followed by Bonferroni post-tests when the ANOVA test indicated significant differences. 

## 3. Results

### 3.1. PRVtmv+ Vaccinated Pigs Remained Clinically Normal and Generated Moderate Levels of CSFV-Specific Neutralizing Antibody Titers

The pigs remained clinically normal after vaccination with PRVtmv+ (intranasal/sub-cutaneous; IN/SQ). On the day of vaccination (0 dpv), mean CSFV-specific neutralizing antibody titers in pigs were less than 1 (negative). On the day of the CSFV challenge (28 dpv), the average CSFV-specific mean neutralizing antibody titers in pigs rose to 25.6 (Figure 1B; Appendix A), which is more than a 25-fold increase (seroconverted). 

### 3.2. PRVtmv+ Vaccinated Pigs Were Protected against Severe CSFV Clinical Disease 

The results revealed that the sham-vaccinated control pigs developed severe CSFV-specific clinical signs, characterized by anorexia, depression, diarrhea, and high fever by 4 dpc, and subsequently became moribund. Therefore, all the sham-vaccinated pigs were euthanized on or before 7 dpc (Figure 2A,B). However, in the PRVtmv+ group, pigs #92 (on 5 dpc) and #84 (on 8 dpc) had a fever of 40.88 °C and 40.22 °C, respectively, for only a day (Figure 2A; Appendix A). Also, beginning on 7 dpc, three vaccinated pigs (#84, 87, and 88) had diarrhea that lasted for several days but recovered. They were otherwise clinically normal until 21 days post challenge, when all the vaccinated pigs were euthanized.

### 3.3. PRVtmv+ Vaccinated Pigs Had a Three-Fold Lower CSFV Viremia in Blood 

Severe forms of CSF in pigs are associated with high levels of viremia. Therefore, we determined the CSFV viremia in pigs following the challenge. Upon virulent CSFV challenge, all sham-vaccinated control pigs developed moderate-to-severe viremia in the blood on 4 dpc, which persisted until their sacrifice (7 dpc), with a mean CSFV titer of 3.4 and 6.05 log_10_ TCID_50_/_mL_, respectively (Figure 2C; Appendix A). In contrast, PRVtmv+ vaccine group pigs developed a lower level of viremia, with a mean CSFV titer of 2.45 and 1.84 log_10_ TCID_50_/_mL_ on 4 and 7 dpc, respectively. Notably, on 7 dpc, the viremia was reduced significantly compared with that of the sham-vaccinated pigs (16,218-fold reduction) and at 14 and 21 (the 21st day being the day of euthanasia) dpc, all PRVtmv+ vaccinated pigs were negative for viremia, based on a sensitive test for the infectious virus particles, and therefore were clear of the virus.

### 3.4. PRVtmv+ Vaccinated Pigs Had Significantly Milder Leukopenia and Thrombocytopenia after the CSFV Challenge 

Sham-vaccinated control group pigs developed severe hematological changes, including leukopenia, lymphocytopenia, and thrombocytopenia, by 4 dpc (Figure 3A–C; Appendix A). The results show that the decrease in leukocyte counts in seven control non-vaccinated pigs was 75% and 80% at 4 and 7 dpc, respectively (Figure 3A; Appendix A). Similarly, the reduction in the control pigs’ platelet counts was approximately 65% on both 4 and 7 days post challenge (Figure 3C; Appendix A). However, the corresponding drop in hematological values in the vaccinated pigs was considerably less; the reduction in the leukocyte counts for the PRVtmv+ vaccinated pigs was 35% and 28% on 4 and 7 dpc, respectively, while the decline in platelet counts for the vaccinated pigs was 38% and 30% on, respectively. Thus, the PRVtmv+ vaccination reduced the immunosuppressive effects of CSFV infection by reducing leukopenia, lymphopenia, and thrombocytopenia levels by at least two-fold relative to the control pigs. 

PRVtmv+ vaccinated pigs were more likely to have a lack of the fatal CSFV disease, the induction of moderate levels of virus-neutralizing titers, a 3-fold reduced viremia, and reduced immunosuppression. This reflected the protective efficacy of the vaccine.

### 3.5. PRVtmv+ Vaccinated Pigs Did Not Have Any CSFV-Specific Gross Lesions upon CSFV Challenge

At 21 days post challenge, the PRVtmv+ vaccinated pigs had no clinical signs when they were euthanized. Notably, no CSFV-specific gross lesions were detectable in the spleen, kidney, and tonsils of the vaccinated pigs (Figure 4). All five moribund sham-vaccinated control pigs, were euthanized on days 6 and 7 post challenge. These results demonstrated that the PRVtmv+ vaccine protected the pigs against the most pronounced CSFV-specific lesions, i.e., infarcts, petechiae/hemorrhages, and necrotic ulcers, seen in the spleen, kidney, and tonsils, respectively, of the CSFV-infected pigs [1]. 

## 4. Discussion

Previously in this paper, we constructed a PRVtmv+ subunit vaccine vector against PCV2b and CSFV. Further, we determined its efficacy against a PCV2b challenge [27], its ability to reactivate from latency, and nasal shedding property upon latency reactivation [35]. The results revealed the following: (i) the PRVtmv+ vaccination protected pigs against the PCV2b challenge by preventing fecal virus shedding, viremia, leukopenia, and lymphocytopenia [27], and (ii) PRVtmv+ caused abortive infection in the trigeminal ganglia (TG) neurons. Therefore, after the latency reactivation of PRVtmv+ in the TG neurons, the virus did not shed in the nasal secretions. Consequently, unlike the wt PRV or traditional modified live PRV vaccines, which are shed in the nasal secretions and maintained and circulated in the pigs, the latent PRVtmv+ vaccine vector cannot spread in the pig population [35]. 

In this study, we demonstrate that PRVtmv+ vaccinated pigs showed a moderate level of CSFV-specific neutralizing antibody titers, and they survived the fatal CSFV challenge until euthanasia at 21 days post challenge. On the contrary, the sham-vaccinated control pigs had to be euthanized by 6–7 dpc because of their moribund conditions. Although some PRVtmv+ vaccinated pigs developed transient fever for a day and mild diarrhea, their condition did not linger beyond a day or two. At euthanasia, 21 days post challenge, all pigs, including one pig with mild diarrhea, were clinically normal. None of the PRVtmv+ vaccinated pigs developed any CSFV-specific gross lesions. Together, these results correlated well and demonstrated that the single PRVtmv+ vaccination was sufficient for pigs to be protected from the fatal consequences of CSFV infection.

The “C”-strain vaccine virus is efficacious against CSFV, but lacks a serological marker. Therefore, the vaccinated animals cannot be distinguished from the infected animals during the sporadic CSFV outbreaks [11]. Due to this limitation, EU countries do not permit the vaccination of pigs with the otherwise highly effective MLV “C”-strain or similar MLV vaccines. Further, there are trade restrictions on the countries using the MLV CSFV or that are not yet CSFV-free, which has enormous economic consequences. 

The PRVtmv+ can be distinguished serologically from the PRV wt- as well as the CSFV wt-infected pigs (DIVA vaccine). Additionally, as noted above, the PRVtmv+ does not replicate in the TG following reactivation from latency. Therefore, the vaccine virus would not circulate in the pig population. As such, it could be used as a safe and emergency vaccine in PRV-free countries to control a CSFV outbreak, i.e., in EU and North American countries. 

The direct and indirect contact of domestic pigs with wild boars has been implicated as a cause of the outbreak of CSFV in Germany Spain, Italy, and the Netherlands [36,37,38]. Thus, CSFV outbreaks in domestic pigs remain a potential threat to the populations of Europe, Central America, and South America, including Brazil and the Caribbean [1,6,8,39]. As noted earlier, routine vaccination against CSFV with the “C”-strain-based vaccination or similar MLVs is not permitted in EU countries because the MLV lacks the DIVA property and cannot be distinguished from the circulating field strain. Instead, a rigorous sero-surveillance of domestic pigs and wild boars has been enforced [40]. 

Previous studies by other investigators revealed that the pre-colonization of TG with a latent PRV prevented the latent infection in the TG by another superinfecting PRV [41]. Even though we did not validate the latter phenomenon in this study, we believe it would also be applicable to PRVtmv+ infected pigs. As noted above, PRVtmv+ established latency but did not replicate in the post-mitotic TG neurons. Consequently, upon dexamethasone-induced latency reactivation, the vaccine virus was not shed in the nasal secretion [35]. Therefore, PRVtmv+ would be suitable as an alternative subunit vaccine against CSFV, even in countries where PRV has been eradicated from the domestic pig population. The potential advantages of using PRVtmv+ as a live subunit vaccine against CSFV are as follows: (i) the PRVtmv+ is highly attenuated for pigs, yet replicates well enough in the nasal mucosa; (ii) it is highly immunogenic and induces CSFV-specific neutralizing antibodies in pigs; (iii) it has the DIVA property; (iv) it stably expresses inserted chimeric proteins; (v) the vector virus establishes latency in the TG neurons but does not replicate, leading to no nasal virus shedding; (vi) it is a bivalent subunit vaccine and protective against both PCV2 and CSFV challenges; and (vii) if needed, it can likely be administered intranasally as a booster vaccine, since the preexisting antibodies do not interfere with the memory immune response when administered intranasally [42,43], which will be validated soon. 

## 5. Conclusions

Our results demonstrated that the PRVtmv+ vaccinated pigs were protected against fatal CSFV disease and survived until euthanasia at 21 days post challenge. Notably, no gross lesions were detected in the vaccinated pigs at necropsy. The PRVtmv+ has the DIVA property against both wild-type PRV and CSFV, and can be used in countries where PRV and CSF have been eradicated from the domestic pig population.

## Figures and Tables

**Figure 2 viruses-15-02143-f002:**
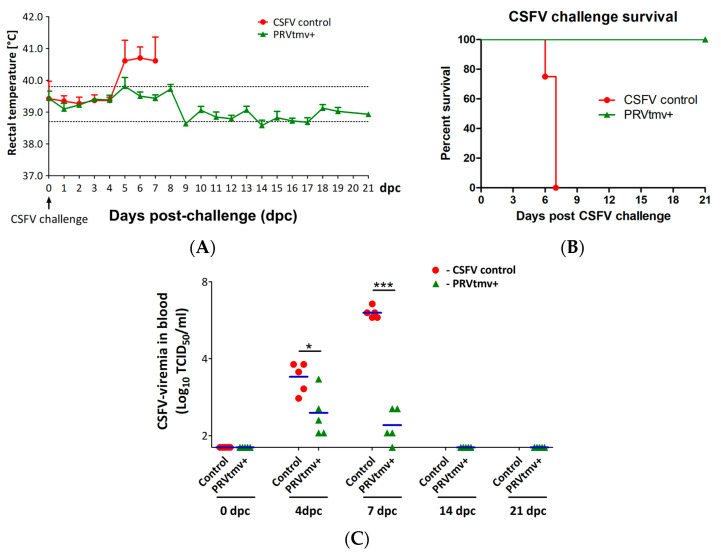
Clinical assessment, survival curve, and classical swine fever virus (CSFV)-associated viremia in sham-vaccinated control and PRVtmv+ vaccinated pigs following CSFV challenge. (**A**) Pigs were challenged with CSFV at 28 days post vaccination (0 days post challenge (dpc)), and rectal temperature was recorded until 21 dpc. Mean rectal temperatures with standard deviation (SD) are given (*n* = 5). All pigs in the CSFV control group were euthanized on days 6 and 7 dpc. (**B**) A survival rate is given in terms of percentage in each group (*n* = 5). (**C**) Blood samples were collected from pigs on days 0, 4, 7, 14, and 21 post challenge, and the CSFV titer was determined in cell culture. The mean CSFV titer in the blood of each animal from both groups is shown. The dot plot graph represents mean + individual values in each group (*n* = 5). TCID50—50% tissue culture infectious dose; * *p* < 0.05 and *** *p* < 0.001.

**Figure 3 viruses-15-02143-f003:**
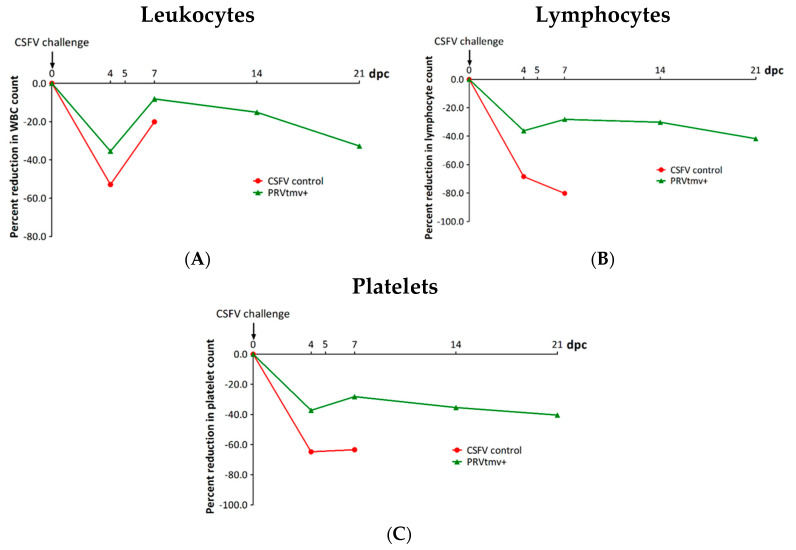
Percent changes in leukocyte, lymphocyte, and platelet count following the classical swine fever virus (CSFV) Brescia strain (BICv) challenge in both groups. Whole blood was collected from pigs on 28 dpv/0 dpc and 49 dpv/21 dpc, (**A**) leukocyte, (**B**) lymphocyte, and (**C**) platelet counts were determined, and their percent changes were calculated. The normal range in pigs: (i) leukocytes—11–22 × 10^3^/μL; (ii) lymphocytes—4.6–12 × 10^3^/μL; (iii) platelets—200–500 × 10^3^/μL [34].

**Figure 4 viruses-15-02143-f004:**
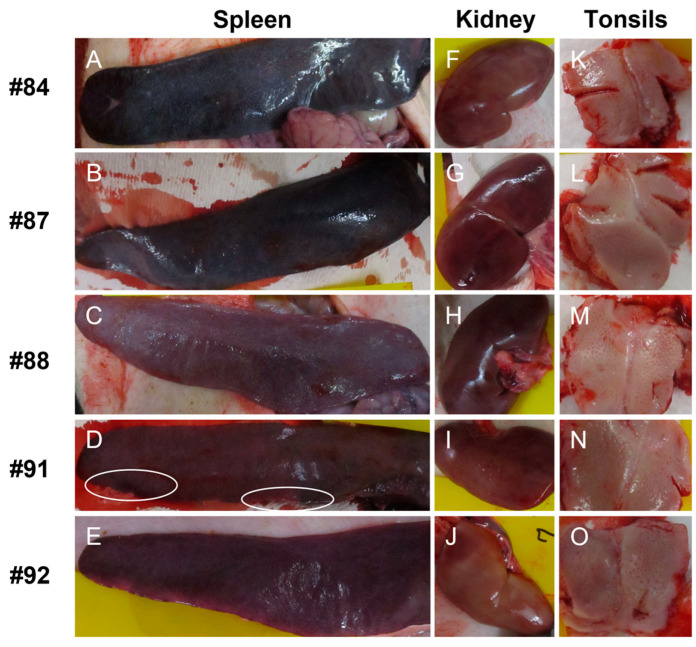
Photograph of spleen, kidney, and tonsils of PRVtmv+ vaccinated pigs (#84, #87, #88, #91, and #92) euthanized at 21 days post challenge. Spleen (**A**–**E**), kidney (**F**–**J**), and tonsils (**K**–**O**). The spleen, kidney, and tonsils lack CSFV-specific lesions. The focal areas marked with circles in D most likely represent areas of blood pooling/incomplete extrusion of blood; this is sometimes seen in the spleens of various species during postmortems after euthanasia using barbiturate compounds. Splenic infarctions, even red-type infarcts, are usually sharply demarcated from the rest of the parenchyma.

## Data Availability

Additional relevant data are available in the Supplementary section of the manuscript.

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
