# Peer review of "A Triple Gene-Deleted Pseudorabies Virus-Vectored Subunit PCV2b and CSFV Vaccine Protect Pigs against a Virulent CSFV Challenge"

_viruses, 2023, doi:10.3390/v15112143_

Round 1
Reviewer 1 Report
Comments and Suggestions for Authors The author should to check some writing details, such as PRV tmv + -Author Response
Response attached

Reviewer 2 Report
Comments and Suggestions for Authors
The paper describes a triple gene-deleted pseudorabies virus-vectored subunit 2 PCV2b and CSFV vaccine protect pigs against virulent CSFV 3 challenges. They developed a safe and effective vaccine against CSF by constructing a triple gene-deleted pseudorabies virus (PRVtmv) vectored bivalent subunit vaccine against porcine circovirus type 2b (PCV2b) and CSFV (PRVtmv+). The results demonstrated that the PRVtmv+ -vaccinated pigs are protected against virulent CSFV challenge as all control pigs got CSFV but not those vaccinated.
Introduction, line 33: I suggest deleting “of” in the sentence “many countries of, including China,”.
Materials and Methods: Clearly and well written. Should be able to reproduce the project with current description.
Results and Figures: Clearly and well written.
Discussion: Well written.
Conclusion: Nice, summary of potential use as DIVA test for emergency vaccination.
Comments on the Quality of English Language
Just check general use of English
Reviewer 3 Report
Comments and Suggestions for Authors
This manuscript reports the results obtained in a trial designed to determine the effectiveness of a recombinant vaccine on protecting pigs from Classical Swine Fever. Results presented in this manuscript complement the results from a previous publication, in which other characteristics of the vaccine, along with the tests on its effectiveness to protect pigs from PCV2 related clinical signs. The manuscript is overall very well written and structured, with minor changes to be addressed, as listed below. Among the modifications suggested, the most important is the inclusion of images/data related to the gross pathology of the non-vaccinated CSFv-challenged pigs that had to be euthanized due to the severity of the clinical signs. Another point of attention is related to some outdated references used and lack of inclusion of important references.
Page 1, line 17 and 22: the way you worded “control group” is difficult to understand. I suggest you add “sham-vaccinated” in parentheses in line 17, thus in line 22 it will be clear that the control groups means that.
Page 1, line38: Check the format of references.
Page 1, lines 38-40: The sonly presence of the virus within a big continent like the America is not a real risk factor per si. I would complement a sentence with some information regarding movement of pigs and people, contaminated food, and others.
Page 2, lines 59-60: Various countries in the World are free of pseudorabies or have only sporadic cases, therefore, the disease is not currently a big economic risk. The reference utilized for this part is too outdated (1999).
Page 2, line 62: Why are those words italicized?
Page 2, lines 62-64: Another outdated reference (2011). Although the facts stated might still be true, to make your point, a more recent reference is needed. Also, add that there is a risk of introduction in free-countries, areas or commercial herds, since in the US, for example, there are reports of recent detection of PRV in wild boars.
Page 2, line 66: Clarify what does PCV2 stands for, since it is the first time this acronym appears in the text.
Page 2, line 68: Did you mean “countries”?
Page 3, line 110: How the healthy status was determined? Clinical evaluation? Any laboratory tests, especially for PCV2?
Page 3: Check the format of the subtitles, as they don’t seem to be consistent.
Page 5, line 190: Did you investigate the cause of the diarrhea? What tests were performed and what were the results?
Page 5, Figure 2: Include a legend for graph C. What do the red triangles represent? And the green circles? Keep the shapes and colors consistent in relation to other graphs.
Page 6, item 3.5: Where are the results from the gross evaluation of the non-vaccinated pigs that had to be euthanized due to the severity of clinical signs? Create a comparative figure containing the images from vaccinated and non-vaccinated pigs for reference.
Page 7, lines 266-268: One of the most important reasons why the available vaccines are not allowed to be used in various countries is not only the risk of shedding live attenuated vaccines, but also the impossibility of differentiated vaccinated from naturally infected pigs, as it was well explained in the introduction. Consider rephrasing this part of the discussion to avoid misleading the readers.
Page 7, line 278: Correct the typo on “direct”.
Page 7, lines 276-280: I suggest you include Brazil in the discussion, as it is one of the biggest pork producers in the world and has an endemic area for CSF. Suggested reference: doi: 10.3390/v12111327
Page 8, line 286: State what TG stands for.
Page 8, Conclusion: Based in your results, you can not determine if the disease due to the CSFv challenge would be fatal, as the moribund pigs were euthanized.
